# Group contracts and sustainability: Experimental evidence from smallholder seed production

**Prakashan Chellattan Veettil**[1]*, **Yashodha**[2]☯, **Judit Johny**[1]☯

1 Impact Evaluation, Foresight and Policy, International Rice Research Institute, Manila, Philippines,
2 Impact Evaluation, Policy & Foresight Unit, International Water Management Institute, Colombo, Srilanka

☯ These authors contributed equally to this work.
* pc.veettil@irri.org

**Data Availability Statement:** The data is available at Harvard data verse: https://doi.org/10.7910/DVN/3OUPKS.

**Funding:** This study is funded by Bill and Melinda Gates Foundation and awarded to the author PCV.

## Abstract

Contract farming in seed production has played an instrumental role in bringing private investment into seed research and production. As developing countries have predominantly small and marginal farmers, the number of inefficiencies that arise from seed contractual agreements hinders producers from realizing the full potential benefits from seed contracts. We carried out an economic experiment with real producers and organizers currently engaged in seed production to analyze their preference for group seed contracts, its sustainability and welfare implications in the seed value chain. The producers are offered two types of group contracts: B and C. Contract B involves a company-organizer-seed producer group (SPG) whereas contract C removes the organizer and directly engages with the SPG (company → SPG). In the experiment, producers are asked to choose between an existing contract and either of the proposed group contracts. The experiment consists of two treatments: (i) concealed and revealed price information between agents, and (ii) presence and absence of a local organizer while making the decision. We find that the preference for group contract B is higher than for group contract C, suggesting the need for producers bargaining which can be achieved through group contract in the existing contract, Bargaining is high (6.3 percentage points) when price information is concealed. SPGs survive for about four out of five rounds and more than half of the groups (53%) formed in the first round survived throughout the five rounds, indicating a very high group sustainability.

## Introduction

Contract farming (CF) in agriculture has diverse views on its impact on smallholders–supporters argue for its inclusivity of smallholders to modern value chains and viewed as an institutional mechanism to remove market imperfections in output, land, credit, labor and insurance markets [1,2]. The critics have highlighted critical issues of CF such as information asymmetry and the imbalance of bargaining power between contractual parties [3–6]. The negative effects are accelerated by the underdeveloped legal institutions and compliance framework leading the smallholders exposed to exploitation and higher risks [7]. Manipulations may occur at

Grant number is OPP1118610. The funders had no role in study design, data collection and analysis, decision to publish, or preparation of the manuscript.

**Competing interests:** The authors have declared that no competing interests exist.

various levels and firms tend to favor large farmers, delay payments, not provide compensation for natural calamity loss, and conceal the pricing method [1,8]. In order to effectively address the limitations of contract farming with smallholders in developing countries, efforts have been made in the past to link farmer groups with agri-business firms [9,10].

Given this context, the current study explores the formation of voluntary farmer groups as an effective institutional strategy to address the problem of asymmetric information and imbalance in bargaining power between agents in contract farming. In particular, we explore producer preferences for the formation of seed producer groups (SPGs) and examine the survival of SPGs over a period of time in Telangana, India.

Borrowing from the existing studies, we expect that voluntary farmer groups, if established based on the product and local context, can benefit agri-business firms and local middlemen as well as farmers in a contract farming setup. Thus, this study has attempted to (a) gather evidence on rice seed producers' preference for voluntary group formation to engage under a contract farming scenario and (b) analyze the sustainability of such groups among rice seed producers. We hypothesize that voluntary group formation by rice seed producers under contract farming, along with price transparency and increased bargaining power for the farmers, can overcome the challenges of asymmetric information and positively impact all agents involved in the contract process.

Ideally, by engaging with producer groups, companies can take advantage of a larger area under the same variety and reduced transaction costs by synchronizing and coordinating activities, and enjoy benefit of scale of operations with each contract. In addition, companies can gain an experienced and loyal producer base by redistributing welfare through minimized rent-seeking behavior of organizers and increased profit. By engaging with producer groups, the local contract organizers (analogous to middlemen) can reduce transaction and compliance costs, and the risks involved in seed production, and can more effectively transfer technology for uniform quality seed production. Further, by being in the producers' group, seed producers can collectively bargain for a higher procurement price and buy inputs and other services in bulk at lower prices. Thus, technically, group contracts reduce the transaction cost and overall costs of seed production, and increase welfare distribution across the seed value chain.

We carried out a lab-in-the-field experiment with real producers and local contract middlemen (who are known as seed *organizers*) in the state of Telangana, India. Organizers are usually business oriented, experienced, influential, and individuals well connected with seed producers and local services. These are individuals from within the locality who identify smallholders willing to undertake seed production and facilitate seed production for agri-business firms or seed companies. In brief, we carried out an experiment with rice seed producers currently engaged in seed production via contract farming with seed companies, under the guidance of a local organizer. In the existing contract (let us call it A), producers have an individual informal contract agreement with the company through an organizer (company → organizer → farmer). The company reveals the price and other contract conditions to a producer through the organizer. The individual producer can either accept the contract at the price given by the organizer or reject the contract. In the experiment, we offered two types of group contracts, B and C. Contract B is similar to the existing type of contract except that producers are given the opportunity to form a seed producer group (SPG) and bargain for the best price with the organizer (company → organizer → SPG). Contract C is similar to B, in which the formation of a SPG and price bargaining are allowed, but the SPG engages in a direct contract with the company without the organizer (company → SPG). Subjects are presented with a choice between the existing contract (A) and one of the group contracts (B or C). Subjects make decisions in five rounds, considering each round as a different production season.

Our study adds some key contributions to the existing literature on contract farming and farmer collectivization. Very few empirical studies exist on contract farming in staple food grains. Studies have largely focused on the factors that determine the success or failure of groups after groups have been established [2,11,12] In this light, the current study empirically evaluates the formation of voluntary groups among rice seed producers under contract farming. Although the possibility of combining contract farming and group formation has been discussed in the literature, little or no empirical evidence indicates the dynamics of group formation. Further, great concerns exist about producers' commitment to organize such groups, coordinate group activities, and ensure no free-riding, which are key factors for success and sustainability. In this study, we evaluate the dynamics of group formation and its potential sustainability over time.

The rest of the study is organized as follows. Section 2 provides a brief background about the Indian seed industry and rice seed production in particular. Section 3 reviews contract literature with a focus on group contracts. Section 4 includes a description of the study location and sampling procedure. Section 5 explains the experimental design and Section 6 presents the empirical strategy, followed by a brief discussion on key results from the experiment in Section 7. We conclude the study in Section 8.

## India's seed industry

A series of policy changes in the 1990s resulted in a competitive seed industry and led to an increase in private investment and R&D in the seed sector [13]. At present, the Indian seed industry is the fifth largest seed market in the world, with a value of USD 2,078.3 million, and is expected to register a compound annual growth rate of 6.4% during 2018–23. However, the impact of these changes has been less in the case of cereals.

**Rice seed industry.** Rice seed production is composed of both public and private firms. Public seed firms operate primarily through progressive farmers who produce improved seed developed by public research organizations and market it through public seed distribution channels [14]. Because of the low-value, low-margin nature of the market and relatively few technologies available to encourage innovation by private firms, this is not a lucrative investment for the private sector [14], especially in the case of open-pollinated varieties. The notable share of private investment in rice happened via rice hybrids. The size of the Indian hybrid rice market during 2008–09 was estimated at about 35,000 metric tons, with a total value of USD 142 million. During 2009–10, the share of private seed companies in certified seed production (hybrid and open-pollinated varieties) of rice was 63.9%. Private seed firms invest considerably in R&D, seed multiplication through farmers, and markets through private distributors and seed dealers' networks.

**Rice seed production.** The majority of rice seed production under the private sector happens via contract farming, in which farmers (henceforth seed "producers") multiply breeder seeds following the practices recommended by the seed companies for a pre-fixed price. Seed production requires companies to provide technical assistance to producers throughout the different stages of production to ensure seed quality. For example, in hybrid seed production, male plant seeds are transplanted a few days earlier than the females to make sure they flower at the same time. Some techniques such as clipping and crossing need to be done to ensure cross-fertilization. Given that the majority of the producers in India are small and marginal farmers, individual contracts with producers incur a high transaction cost for companies. It stands to reason that the seed companies resort to sub-contracting with a local middleman (henceforth "organizer"), who organizes producers, delivers breeder seeds and technical inputs required to the farmers, as well as oversees the entire seed production process to meet quality

standards. All information, including the procurement price, quality standards, quantity requirements, and commission rates per kg of seeds for the organizers, is communicated by the company at the beginning of the season and the two parties enter into a contractual agreement.

Seed producers are selected by organizers based on certain key characteristics: location, land size, soil type, financial capacity, and availability of irrigation. The contractual agreement between producers and the company is instituted by the organizer, and is largely verbal in nature (although a couple of multinational companies engaged in hybrid seed production have written contracts directly with producers, they are not legalized. Seed producers rely heavily on organizers and their explanations of the conditions in those contracts that are written in English). All contractual conditions, including price, are communicated to the producers via organizers. The producers are largely dependent on the organizers: in several cases, growers tend to choose a company based on the organizer.

**Concerns.** Producers' overdependence on organizers in the current seed contract structure not only hinders direct communication between producers and the company but also aggravates asymmetric information, leading to exploitation of producer surplus. In other words, producers are unaware of the actual procurement prices and the organizer commission rate offered by the company. This information asymmetry between organizers and producers might result in price manipulation and a differential pricing strategy by organizers to extract more rent from producers. Moreover, in the current contract structure, producers have no bargaining power with the company. As Little and Watts [15] affirm, when the balance of power between the contracting parties is large, contract farming leads to exploitation of producers. Thus, in this study, we have attempted to introduce price transparency and increase the bargaining power of seed producers to overcome the challenges of asymmetric information.

## Literature review

Contract farming (CF) is usually referred to an agreement between a buyer (seed company) and the farmers (seed producers), that stipulates the conditions on volume, quality, timing of delivery of product, use of inputs, and price or pricing formula [16–19]. Broadly, two types of contracts are prevalent in farming -production contract and marketing contract [4,20]. Under production contracts farmers lose the autonomy or production where contractor influences the production by providing technical assistance and inputs in order to get agreed quantity and quality of product, whereas in marketing contracts the agreement is only on the quantity and quality of the product at a future date at a pre-determined price or pricing formula [21]. Seed production in the study context mostly belong to the production contracts where seed companies stipulates the smallholder seed producers on production of quality seeds of desired quantity mostly implemented through organizer. Seed company supplies the raw material for the seed production (foundation seeds or male and female lines for hybrid seeds), provide technical assistance and organizer facilitates the production through improved credit and input services.

The role of contract farming (CF) in agriculture has been a much-debated topic over the years [7,19,22–25]. Proponents of CF argue that it has the means to incorporate smallholders into the modern sector (commercial and high value agriculture) and is viewed as an institutional mechanism that can remove market imperfections in output, credit, land, labor, and insurance markets [1,2] and address the related constraints faced by smallholders. That is, smallholders through contract farming can enjoy the advantages of improved access to credit, better production methods, and risk capacity (usually attributed to large scale production) as

well as utilize their strength in small-scale production such as family labor and high motivation [26]. Field evidence has shown that CF ensures guaranteed prices and reduced transaction costs through vertical coordination, which results in better quality of output, increased income, and transfer of technology [27–31].

On the other hand, several studies have discussed the negative impacts of CF [8,17,32]. Information asymmetry and the imbalance of bargaining power between contractual parties are critical issues in the CF literature [3–6]. Further, the lack of a proper legal and compliance framework to support new institutional mechanisms in many developing countries can leave smallholders exposed to risks [7]. In the absence of a strong legal framework, companies and other agents in the seed production chain could falsify contracts by raising quality standards to control volume, alter the agreed prices, or even resort to cheating [5,17,33]. Manipulations may occur at various levels and firms tend to favor large farmers, delay payments, not provide compensation for natural calamity loss, and conceal the pricing method [1,8]. Because of the small holdings of producers in developing countries, contracting firms in general do not directly engage with individual producers due to the higher transaction cost. Instead, firms informally use local personnel to organize the contract (organizer) by engaging local producers to reduce the transaction cost. This further increments the asymmetric information problem and probability of a producer surplus.

In order to effectively address the limitations of contract farming with smallholders in developing countries, efforts have been made in the past to link farmer groups with agri-business firms [9]. Coulter et al. [10] suggest contractually linking small, cohesive "linkage-dependent" farmer groups with agri-business firms in order to address concerns of farmer defaults and high transaction costs of supervision and technology transfer incurred by the companies, and to provide increased negotiating strength for the farmers. However, the formation of successful and sustainable farmer groups is dependent on the product and context. For example, Roy and Thorat [34] showed that, in India, marketing cooperatives for grapes reduced transaction costs and contributed to a better bargaining position of smallholders. Coffee cooperatives in Costa Rica facilitated small-scale growers' participation in specialty markets and gained higher prices for their product [35]. Witcombe et al. [36] successfully established sustainable seed producer groups in Chitwan District of Nepal by giving emphasis to strengthening the groups' marketing and managerial capabilities rather than following the common approach of providing training in technical issues.

At the same time, the prevalence of numerous failures in collective action has been a recurring discussion in the literature [37,38]. Farmer groups often struggle with the lack of capital to grow in scale, management capacity, and good organizational governance [39]. In recent times, the promotion of farmer organizations through outside assistance has also re-gained popularity in the context of agri-food system transformation (e.g., [40,41]). However, Markelova et al. [37] point out that the capacity of groups to develop their own rules, rather than just following externally imposed rules, is important for group sustainability. Although some degree of outside assistance (both financial and in terms of capacity building) is often required for producer groups to form and operate successfully, the balance between sustainability and dependency of the group is crucial. Thus, a synthesis of past studies indicates that no "one size fits all" strategy exists to ensure successful and sustainable collective enterprises.

## Materials and methods

### Study area and sampling

As the purpose of the study was to understand producers' preference for and dynamics of a group seed contract, the study area was chosen based on the intensity of rice seed production.

The states of Andhra Pradesh and Telangana are commonly known as the Rice Seed Hub of India. These are the states where the private seed industry first began its operations in the 1980s, concentrating on hybrid pearl millet and sorghum [42]. This made it easier for private firms to expand their business into rice and position themselves for the growing hybrid rice seed market. This region is also known for its suitable climatic conditions and assured irrigation facilities, particularly for rice seed production. In addition, the region offers a concentration of experienced seed growers as well as processing and storage facilities. This study was conducted in the districts of Karimnagar and Warangal of Telangana. These districts have a large presence of the private sector involving many of the major local, national, and multinational players of the seed industry [43].

Based on a series of discussions with stakeholders from the state (Department of Agriculture, seed cooperatives, seed certification agencies, and researchers from the agricultural university), we identified several villages from the districts of Karimnagar and Warangal of Telangana. Sixty villages were selected based on the intensity of seed production and the concentration of production of hybrid varieties and open-pollinated varieties (OPVs). A short village profile survey was done in the selected villages to profile the producers and number of companies and organizers operating in the village. From each village, an organizer and 10–12 producers were selected randomly from the village profile survey to carry out a detailed seed contract and production profile survey. From the total profiled rice seed producers, we randomly and proportionally selected producers to represent production of hybrids and open-pollinated varieties. The surveys were administered to selected producers and organizers in June and July in 2017 to understand the existing seed contract arrangements from both parties. The final sample includes 593 seed producers and 60 seed organizers.

**Characteristics of seed contracts.** The contract characteristics observed for rabi 2017 rice seed production in the study area are presented in Table 1. National and multinational companies dominate hybrid seed production, whereas local companies invest in OPVs. In both OPV and hybrid seed production, very few producers engage in direct contact with the companies. Most payments are made in cash through the organizer within one to two months after seed delivery. On average, producers received INR 64 and INR 15 per kg of seeds, respectively, for hybrids and OPVs. Organizers interact more frequently (every week) with producers in order to technically guide them. Less than 15% of the producers experienced seed rejection in the previous season.

Further, the socioeconomic characteristics of producers and organizers indicate that the organizers are younger (41 years) than producers (46 years), better educated (3.45 years higher), and both of them have fairly good experience (>10 years) in seed production (see Table A1 in S1 Appendix). We observed high variation in seed yield, with an average productivity of about 9 and 24 quintals per acre under hybrids and OPVs, respectively. We did not find any statistical difference on socioeconomic characteristics between hybrid and OPV seed producers. On average, companies offer a commission rate of INR 1.5 per kg and INR 3.0 per kg of OPV and hybrid seeds supplied, respectively.

## Experiment

A lab-in-the-field experiment was carried out in September of 2017 with the same producers and organizers who participated in our survey. These producers and organizers were approached independently and simultaneously during the experiment. The experiment was carried out using a pre-tested protocol.

**Experimental design.** Our design involves three types of rice seed contracts. Let us call them A, B, and C. Contract A represents the existing contract structure, which involves a

**Table 1. Seed contract characteristics in the study area.**

| S. no. | Particulars | Hybrid | OPV |
|---|---|---|---|
| 1 | Direct contact with company | 0.10 | 0.28 |
| 2 | *Contracted company* | | |
| | Cooperative and others | 0.01 | 0.37 |
| | Multinational | 0.58 | 0.00 |
| | National | 0.35 | 0.12 |
| | Local | 0.07 | 0.50 |
| 3 | Written agreement | 0.27 | 0.14 |
| 4 | *Payment* | | |
| 4.1 | Price per kg | 64.00 (9.41) | 15.78 (1.03) |
| 4.2 | *Payment through* | | |
| | Company | 0.19 | 0.47 |
| | Organizer | 0.80 | 0.53 |
| 4.3 | *Mode of payment* | | |
| | Cash | 0.60 | 0.47 |
| | Bank transfer | 0.40 | 0.53 |
| 4.4 | *Payment waiting time* | | |
| | Less than a month | 0.14 | 0.25 |
| | 1–2 months | 0.70 | 0.55 |
| | More than 2 months | 0.16 | 0.20 |
| 4.5 | *Organizer interaction* | | |
| | Weekly | 0.44 | 0.36 |
| | Alternate weeks | 0.38 | 0.34 |
| | Monthly | 0.11 | 0.20 |
| | Alternate months | 0.07 | 0.09 |
| 5. | Rejection experienced | 0.14 | 0.13 |
| No. of observations[a] | | 278 | 303 |

Characteristics are expressed in terms of proportion. In parentheses are standard deviations.

[a]Out of 596 producers, 15 did not know the seed produced was OPV or hybrid.

company, organizer, and individual producer (company → organizer → producer). Contract B consists of a company, organizer, and seed producer group (SPG), where the organizer engages with the SPG instead of individual producers (company → organizer → SPG). Contract C consists of a company and SPG; the organizer does not have a role; instead, the company interacts directly with the SPG (company → SPG). Contracts B and C are group contracts, which involve collective decision making by the producers at various levels.

For the experiment, once the producers arrived, they were welcomed and gathered into a community hall identified for the purpose in the respective villages. The producers of a village were matched with an organizer (previously identified) from another village without revealing each other's identity. Likewise, each organizer was matched with 10–12 producers from a different village than his. For the organizer, the experiment was carried out at his house/office. Concealing the identity of producers and organizers is important since both parties may have interactions in the future, which might affect their decision in the experiment, anticipating post-experimental punishment. Detailed explanations about the characteristics and structure of contracts A, B, and C were provided to all participants. They were also informed that the money they earn depends on their decisions as well as the matched organizer's decision. Therefore, they were asked to make decisions seriously. An endowment of INR 300 was given

to each producer. A matching endowment of INR 300 per producer was given to organizer and the total endowment of the organizer depends on the number of producers s/he is matched with. Both participants were given the same role as they had in their current seed contract.

The experiment has four stages: (i) contract choice, (ii) price agreement, (iii) production decision, and (iv) payout stage.

*i) Contract choice*: In this stage, seed producers were given a choice between individual contract A and one of the group contracts (B or C). The choice between A and B or A and C was assigned randomly at the village level, that is, within the village, all subjects faced similar paired choices. Our interest is to explore voluntary group formation in the existing institutional, socioeconomic, and political context. Since the focus is on group formation within the village, introducing more than one type of group contract within the village is not feasible. Further, practically, it was difficult to randomize at the producer level due to contamination of treatments. The producers who chose contract A proceeded through the next stages individually. If either contract B or C was chosen, producers were asked to form an SPG. The main criteria to form an SPG were (a) minimum membership of five producers and (b) an individual membership fee of INR 50 per season. In the case of a group contract, members of the SPG jointly enter the next stage of the experiment. If the producers who chose B or C were not able to come together and form a group to meet the minimum criteria, then they were asked to continue with contract A. Producers were also given a choice to opt out if they preferred not to produce in that season.

*ii) Price agreement*: In this stage, the procurement price of seed per kg along with other terms and conditions of production were communicated by the organizers. In contracts A and B, producers are offered the price by the organizer. In contract A, individual producers could choose to produce for the price specified by the organizer or opt out of production if they did not agree with the price. In contract B, the members of the SPG could either collectively accept the price offered by the organizer or bargain for a higher price. The number of bargaining turns between parties was capped at three. If the price was not settled after three chances of negotiation, the members of the SPG had to accept the initial price quoted by the organizer and the organizer was fined INR 50. This ensured that both parties accounted for the costs of negotiation failure.

In contract C, the price and contract agreements were communicated by the company to the SPG. The SPG was given the option to either accept the price quoted by the company or bargain for a higher price, that is, producers can quote from a minimum of INR 1 to a maximum of INR 5 per kg of seed above the company-offered price. The price negotiation rounds were capped at three. However, unlike in A or B, the company acceptance or rejection of a price in the bargaining process had some probability distribution in C. The probability that the company accepts the additional price of INR 1, INR 2, INR 3, INR 4, and INR 5 quoted by the SPG was 0.9%, 0.7%, 0.3%, 0.1%, and less than 0.1%, respectively. If the company rejected the price quoted by the SPG in the negotiations, SPG had to accept the first price offered by the company.

The seed companies did not become involved in the experiment directly as sealed envelopes were used on their behalf. This is to avoid post-experimental friction as well as not to reveal the identity of the company to producers and organizers and vice-versa. The envelopes consisted of information regarding seed price, commission rates, and accepted quality standards. In contracts A and B, the experimenter handed over the sealed envelope to the organizer. The communication between organizers and producers about the price agreement happened over an audio call. The experimenter recorded the price offered by the organizer. In the case of C, the sealed envelope was opened by the experimenter in front of the producers and read out.

Producers who chose individual contracts were separated from those who chose group contracts. Further, if more than one group was formed in a village, then the groups were seated separately for the remaining stages of the experiment. In the case of a group contract, the members were asked to choose one person to communicate on behalf of the group to the organizer over the phone. Care was taken to restrict all communication to price negotiations. Subjects were instructed not to reveal information about their identity or location or any other personal details.

*iii) Production*: Production requires efforts from both the producer and organizer, which may affect both the quantity and quality of production. To make the decision simple, we assumed that the intensity of efforts by producers and organizers affects the quality of production, but not the quantity. To capture the effort involved in the production process, we introduced different levels of effort and their associated costs. Producers and organizers were given the option to choose the level of effort that they wanted to invest in the production process. The minimum effort level was E1, which cost INR 50, and the maximum effort level was E5, which cost INR 250. The cost of each level of effort was the same for both producers and organizers. The effort decisions were made simultaneously and independently by producers and organizers regardless of the type of contract and were not revealed to each other.

Further, seed production faces two types of risk: production and weather risk. Suitable weather conditions are critical for successful seed production [44,45]. Weather risk is exogenous. We assumed that there is a 10% chance that bad weather may wipe out the output, leading to a no-product scenario in all three contracts. Production risk is endogenous to effort level, that is, production risk decreases with an increase in effort added into production. In contracts A and B, the organizer's efforts depict the efforts he/she would make to provide technical guidance to producers, and thus assumed no production risk in A and B. In contract C, there is no organizer to provide technical guidelines; thus, producers face production risk in addition to weather risk. The probability of facing a production risk increases with decreasing effort levels. Unlike in weather risk, facing production risk reduces the quality of seed produced by one grade. Producers face production risk first, followed by weather risk (see Table A3 in S1 Appendix).

The probability of weather and production risks was explained and then demonstrated by the experimenter. We used green and red balls to represent good and bad weather or production risk, respectively. For a given probability of weather or production risk, an appropriate number of green and red balls was placed in a bag and subjects were asked to pick a ball. Irrespective of the type of contract chosen, subjects faced production or weather risk individually.

*iv) Output quality*: There are three quality standards, sub-standard (Q3), standard (Q2), and excellent quality (Q1), that producers can produce. The company accepted quality levels Q2 and Q1 and received the agreed price; minimum acceptable quality level Q2 in order to obtain the agreed price; and, if the quality is Q1, producers obtain 10% more of an agreed price as an incentive for higher quality seeds. Sub-standard quality of seeds will be rejected by the company. In case of rejection, producers are free to sell the product in the grain market for a fixed amount of INR 100.

The final quality of seeds was ascertained after producers and organizers chose their respective effort levels and event of risk, if any. Table 2 represents the effort levels corresponding to different quality standards under good weather conditions in the case of contracts A and B. A notable feature is that if one party chooses higher effort and the other party chooses lower effort, the chances of achieving the acceptable seed quality are lower. If both parties choose higher effort levels, the chances of achieving the standard and excellent quality are higher.

In the case of contracts B and C, the final quality of the seed lot produced by the SPG as a group was taken into account. Final group quality is estimated in two steps. First, quality is

**Table 2. Effort levels of producers and organizers corresponding to different quality under good weather in contracts A and B.**

| Producer | | Organizer | | | | | |
|---|---|---|---|---|---|---|---|
| | | | OE1 | OE2 | OE3 | OE4 | OE5 |
| | Effort | Effort Costs (INR) | 50 | 100 | 150 | 200 | 250 |
| | PE1 | 50 | Q3 | Q3 | Q3 | Q3 | Q2 |
| | PE2 | 100 | Q3 | Q3 | Q3 | Q2 | Q2 |
| | PE3 | 150 | Q3 | Q3 | Q2 | Q1 | Q1 |
| | PE4 | 200 | Q3 | Q2 | Q1 | Q1 | Q1 |
| | PE5 | 250 | Q2 | Q2 | Q1 | Q1 | Q1 |

Q1 = excellent quality, Q2 = standard quality, and Q3 = sub-standard quality.

determined individually as in the case of A for each member of the SPG for his/her respective plots considering the respective agents' efforts and the risk faced. In contract B, the quality depends on the producer's effort level, organizer's effort level, and the weather risk faced by the producer. In contract C, the individual quality depends on the producer's effort level, production risk, and weather risk. For contract C, there are four potential possibilities that the producer might have faced here: (i) faced no production risk and no weather risk, (ii) faced only production risk and no weather risk, (iii) faced no production risk and only weather risk, and (iv) faced both production risk and weather risk. In cases (iii) and (iv), no output was observed due to weather risk. In cases (i) and (ii), output will be observed, but the quality depends on the effort level chosen by the producer. Second, to calculate seed quality of the SPG, the individual qualities achieved by the SPG members are aggregated using the weighted average method. The weights used are 0, 0.5, and 1 for quality Q3, Q2, and Q1, respectively. If the average factor is less than 0.5, the quality assigned is sub-standard (Q3); from 0.5 to 0.75, the aggregated group quality is standard (Q2); and, if it is greater than 0.75, the accomplished aggregated group quality is excellent (Q1). If a member faces a weather risk in the SPG, the quality of seeds produced by the remaining members in the SPG is aggregated. Hence, in group contracts, the final group quality is a result of individual effort decisions, risks faced, the organizer's effort decision, as well as the risk faced by and effort decisions of other group members. The calculated average seed quality of the SPG was informed to all members at the end of the production stage in both B and C contracts without revealing individual seed quality and effort invested, accounting also for traditional free-rider issues in the system.

Producers and organizers follow these sequences of decisions for five rounds. Each round was asked to be considered as a different season and thus the price of seeds could differ according to market demand. Area and productivity were kept constant to make the decision simple. The company provided five envelopes with varied price and a randomly drawn envelop was used each round. The productivity assumed for OPVs and hybrids is 20 and 9 quintals per acre, respectively. The envelope consists of INR 19 and INR 20 per kg of OPVs and INR 63 and INR 65 per kg of hybrids, and the company assigned this randomly into the envelopes. In each round, the envelopes were drawn randomly. The final payoff for producers was calculated after accounting for membership cost, effort cost, price agreed, and quality produced. For the organizer, the payoff was calculated for each producer he/she was matched with. The organizer reveals to the experimenter the price and commission rate mentioned in the envelope after completing all the rounds. The organizer payoff was calculated considering commission rate, effort cost, quality of seeds, and by taking the difference between the price in the envelope and the price offered to the producer. Out of the five rounds, one round was randomly selected for payoff for both the organizer and producer. The final payment of the organizer is an aggregate

of payoff earned with each matched producer for the selected round. The experimenter who is assigned the role of the company representative visits the organizer after concluding the sessions and pays what she/he earns.

**Treatments.** The experiment has two main treatments. In the first treatment, a randomly selected local organizer who is familiar to producers from the village was present during the experiment sessions. The local organizer was neither allowed to communicate with the producers nor was informed about the decision made by the producers. The presence and absence of the local organizer during the producers' choice of the contract is randomized at the village level. Any interventions that aim to improve producers' bargaining power might not be appreciated by the organizer. Given the social and political influence of the organizer in the locality, we hypothesize that the organizer might influence producers in possible means to discourage their decision to participate in the formation of an SPG. This treatment helps to understand such effects, if any, on the choice of the group contract.

The second treatment is called the price information treatment, in which the true price information is either concealed or revealed between the organizer and producers. The concealed case represents the status quo, where the producers do not know the original price and commission rate offered by the company, while the organizer knows them. In the revealed case, the price offered by the company is known to both the organizer and the producers, that is, in the experiment, the company sends out two envelopes, one each for the producers and organizer regarding the price. In the producers' session, the experimenter reads out the price quoted by the company. This treatment aims to understand the price-holding behavior of organizers in the present structure of seed contracts. We randomize the concealed and revealed price information at the village level. For the organizer, the randomization is at the individual level. Table 3 presents the number of subjects and sessions under each treatment. Both treatments apply in the case of A vs. B choices; however, in the case of A vs. C choices, only treatment 1 applies.

We carried out a balance test to compare the socioeconomic characteristics of producers under different treatments. We found no statistical difference in the age, education, landholding, proportion of land devoted to seed production, years of experience in seed production, and variety grown among the producers of different treatments. This ensures that the treatment effects are not contaminated due to existing differences between producers assigned to treatments (see Table A2 in S1 Appendix).

## Ethics statement

The study was approved by Project Management Office (PMO) of International Rice Research Institute (IRRI) [approval number: A-2014-95 (DRPC2014-102)]. We have also taken written consensus from all participants and were informed about the data privacy. Since the study

**Table 3. Number of subjects and experiment sessions under different treatments.**

| S. no. | Particular | Treatments | Paired contract choice faced | |
|---|---|---|---|---|
| | | | A vs. B | A vs. C |
| 1 | Price information | Concealed | 175 (17) | 296 (30) |
| | | Revealed | 125 (13) | |
| 2 | Presence of local organizer | Presence of organizer | 160 (16) | 145 (15) |
| | | Absence of organizer | 140 (14) | 151 (15) |
| Total | | | 300 | 296 |

In parentheses are the number of sessions.

required only the choices of participants in different seed production contracts, no collection of plant, animal, or other materials is required. Further, in order to avoid post experiment conflicts, we implemented the study by matching seed producers with organizer at different locations maintaining anonymity of identities throughout the experiment.

## Empirical strategy

The experimental data are strategized to analyze three aspects of a group contract: first, to estimate the probability of producers' choice of group contract; second, to analyze the determinants and dynamics of group formation in the village using aggregated choices at the village level; and third, to predict the survival rate of the group using the hazard function approach.

**Preference for group contract.** In the choice stage, the producers made a choice between an individual contract and a group contract in each round. The probability of producer $i$ choosing a group contract (either B or C) over A in each round $r$ is estimated as follows:

$$\Pr(Grp\_cont)_{ir} = \alpha + \beta_1(Avs.C)_{ir} + \beta_2(Organizer)_{ir} + \beta_3(Avs.C*Organizer)_{ir} + \delta(Z)_{i(r-1)}$$
$$+ \gamma X_i + \varepsilon_{ir} \tag{1}$$

The outcome variable ($Grp\_cont$) is a choice group contract; it takes 1 if the producer's choice of a contract is either B or C, and zero otherwise. The producers were randomly assigned either A vs. B or A vs. C and asked to make a contract choice between an individual contract (A) and a group contract (B in A vs B or C in A vs C). In treatment 1, they make a contract choice with the presence or absence of an organizer during the choice stage. Our parameter of interest $\beta_1$ measures the likely effect of offering A vs. C contract combination over A vs. B on the producers' choice of group contract. $\beta_2$ measures the likely effect of the presence of an organizer on the producers' choice of group contract. Interaction coefficient $\beta_3$ measures the difference in the likelihood of choice of group contract between A vs. B and A vs. C contracts in the presence and absence of the organizer. The term $Z_{i(r-1)}$ represents the vector of dynamic variables from the previous round ($r$-$1$) for producer $i$ such as if a group contract was chosen, group earnings, and price gain in round $r$-$1$. The coefficient $\delta$ is the vector of parameters to be estimated correspondingly for $Z_{i(r-1)}$. The producers' socioeconomic characteristics, which affect the choice of the group contract, are represented by the vector $X_i$. To estimate (1), we use the individual producer's choice of contract. As the nature of the outcome variable is binary, we employ the panel probit model for estimation.

**Dynamics of group formation.** Group formation at the village level might potentially be influenced by village factors such as social, economic, and political conditions; trust; and experiences among the producers in the village. Understanding the village-level determinants of the formation of an SPG is a key for policymakers to employ strategic intervention to overcome hurdles, if any exist. We estimate group formation as follows:

$$\Pr(Grp\_form)_{vr}$$
$$= \alpha + \beta_1(Avs.C)_{vr} + \beta_2(Organizer)_{vr} + \beta_3(Avs.C*Organizer)_{vr} + \delta(Z)_{V(r-1)} + \gamma W_v + \phi X_v$$
$$+ \varepsilon_{vr} \tag{2}$$

where $Grp\_form$ is an outcome variable denoting whether village $v$ has formed an SPG in round $r$. It takes value 1 if village $v$ has formed at least one group and zero otherwise. $\beta_1$, $\beta_2$, and $\beta_3$ are parameters of treatment effects as mentioned in Section 4.1, which are to be estimated. $Z_{V(r-1)}$ is a vector of dynamic variables from the previous round ($r$-$1$) for village $v$ such as whether a group formed, average price gain, average group earnings of producers, and if members faced rejection in the round ($r$-$1$). The coefficient $\delta$ is the vector of parameters to be

estimated correspondingly for the vector $Z_{V(r-1)}$. $W_v$ represents the vectors of village characteristics such as seed production area, the number of hybrid producers, caste composition, and number of seed companies operating in the village. Further, we also control for the vector of covariate $X_v$ consisting of socioeconomic characteristics of producers in the village.

**Group survival.** The survival rate of the group essentially indicates the sustainability of the SPG. Any survival analysis is generally described in the form of either a survival function or hazard function. The group survival function represents the probability of the SPG surviving from the round of formation to some round beyond $r$ [46]. It describes the survival rate of the group and is usually estimated by the Kaplan-Meier (K-M) survival method. The hazard function gives the instantaneous potential of having an event of group break at round $r$, given the group survival up to that round [46]. The hazard function is mainly used as a diagnostic tool to specify the multivariate model for the survival function to determine the covariate effects (specific challenges relating to estimating the survival rate arise mainly from the fact that when few SPGs experience a break, the survival times will be unknown for the subset of SPGs that survived throughout the rounds (censoring). A simple average in censored data leads to higher estimates of survival rate. Therefore, we used the method that can account for the censoring).

We used the non-parametric K-M survival estimates to analyze the survival rate using observed survival time. Suppose that $k$ SPGs have an event of group break in the rounds of follow-up at distinct rounds $r_1 < r_2 < r_3 < r_4 < r_5. \ldots < r_k$. As an event of group break is assumed to occur independently of one another, the probabilities of an SPG surviving from one round to the next can be multiplied together to give the cumulative survival probability [46]. Therefore, the probability of an SPG surviving in round $r_j$, $S(r_j)$, is calculated from $S(r_{j-1})$; the probability of an SPG surviving at $r_{j-1}$, $n_j$; the number of SPGs surviving just before $r_j$; and $d_j$, the number of SPGs broken at $r_j$, by

$$S(r_j) = S(r_{j-1})\left(1 - \frac{d_j}{n_j}\right) \tag{3}$$

Eq 3 is a step function that changes the value in each round $r$. This function allows each SPG to contribute to the estimates as long as each survives. If all SPGs face a break at some round, this function decreases to the ratio of the number of SPGs facing a break in round $r$ divided by the number of SPGs formed in round one, that is, the survival rate is zero. The K-M survival curve (plot of survival rate against round) provides a useful summary and median estimates of survival rates indicating the sustainability of SPGs.

Further to understanding the determinants of group survival probabilities, we use a Cox proportional hazard model. Following the multivariate approach, it describes the relationship between group break incidence (hazard function) and a set of covariates [47]. The Cox model is mathematically specified as

$$h(t) = h_0(r) \times \exp\{b_1 x_1 + b_2 x_2 + \ldots \ldots + b_p x_p\} \tag{4}$$

where $h(t)$ hazard function depended on a set of $p$ covariates $(x_1, x_2 \ldots x_p)$ whose effect size is measured by $(\beta_1, \beta_2 \ldots \beta_p)$ parameters, respectively. $h_0$ measures the base hazard rate in round zero and it is 1 if all the covariates take value zero. The quantities $\exp(\beta_p)$ are hazard ratios. If the $\exp(\beta_p)$ value is greater than 1, it indicates that, as the value of the $p^{th}$ covariate increases, the group's hazard rate increases and, in other words, decreases the length of group survival.

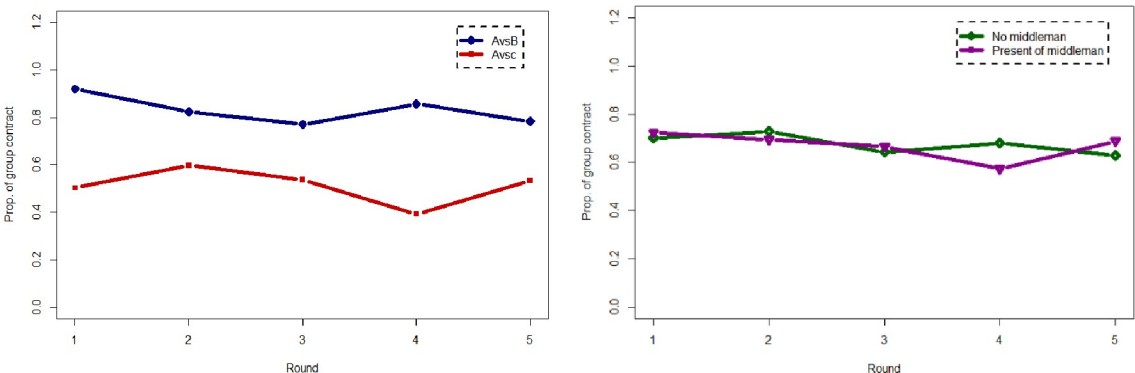

**Fig 1.** Choice of group contract under different contract offer (left) and in the presence of an organizer (right).

## Results and discussion

### Choice of group contract

The proportion of producers choosing a group contract over the rounds is depicted in Fig 1A in S1 Appendix. A total of 85% of the producers chose contract B over A when offered A vs. B contracts, whereas only 50% of the producers chose C over A when offered A vs. C contracts. This suggests that the producers were indifferent between group contract C and A, but prefer contract B to C indicated by a statistically significant proportional difference between A vs. B and A vs. C ($p < 0.05$). The proportion of choice of group contract does not significantly differs with presence and absence of organizer in all rounds, except in the round 4 (Fig 1B).

The sequences of decisions and outcomes after the contract choice stage in the experiment are presented in Table A4 in S1 Appendix. The *Offered price* is the first price offered to the producers either by the organizer or the company. The price offered by the company in contract C is significantly higher than the price offered in contracts A and B, which signals the price manipulation behavior of the middleman. However, the offered price is higher in contract A than in contract B. This suggests that the organizer anticipates bargaining in group contract B and thus offers a lower price than in A. This is further revealed by the *Price gain*, which is the difference between the final settled price and first offered price. The price gain in B is INR 2.0 and INR 1.5 per kg for hybrids and OPVs, respectively, suggesting that bargaining has increased the final price in contract B vis-à-vis contract A. Once producers contest the offered price in B, they use all three rounds to bargain for a higher price. However, the price gain in B is not high enough to match the price offered by the company in C, which indicates that the organizer exerts dominance in the bargaining by offering a lower price in the first place, thus managing to achieve some price margin. Instructors administered the phone conversations between the organizer and producers. The frequent conversations recorded indicate that organizers mention that the difference in price they offered and final price is part of their commission. As producers do not know about organizers' commission charges for service from the company, the organizer exploits the asymmetric information to convince producers while hiding the contractual arrangements between the company and the organizer.

The producers' effort is higher in the case of C, followed by A and B; however, the difference is not statistically significant. The producers' effort level increases over the rounds in group contracts, while it decreases for the organizer. This indicates the strategic behavior of the organizer. Overall, the producers' earnings are highest in contract B (INR 693), followed by A (INR 646) and C (INR 529). Earnings are lowest under contract C, owing to the high production risk results in low quality seed (see Fig A2 in S1 Appendix). Nevertheless, over the rounds,

earnings in C increase as production risk decreases through increased effort levels. Organizer earnings per producer are higher in contract A (INR 72) than in B (INR 63) after accounting for *price margin* (envelop price – final price).

## Preference for group contract

To establish a link between producers' contract choice over treatment and their actual contract experiences, we estimated Eq (1) using a panel probit model. The estimated average marginal effects are presented in Table 4. Conditional on the relevant characteristics of the producers, the estimates from columns 1 to 3 indicate that the likelihood of choice of group contract is significantly higher when offered A vs. B than A vs. C contract. The marginal effect implies that the choice of group contract B is 24 percentage points more likely than the choice of group contract C. The estimates of treatment 2 suggest that the presence of an organizer during the choice stage significantly reduces producers' preference for a group contract. This result implies that the organizer has high power to adversely influence the choices of producers, that is, merely by his/her presence, the choice of a group contract decreases by 10 percentage points.

The interaction effects indicate the difference in the likelihood of choice of group contract between A vs. B and A vs. C in the presence and absence of an organizer, that is, (A vs. B * absence organizer – A vs. C * absence organizer) – (A vs. B * presence organizer – A vs. C * presence organizer). We derived the average marginal effects for all treatment interactions. Average marginal effects of A vs. B * absence of middleman is 0.8058 (SE 0.0320), A vs. B * middleman present is 0.7383 (SE 0.0262), A vs. C * no middleman is 0.5699 (SE 0.0238), and A vs. C* middleman present is 0.6394 (SE 0.0236). The predicted margins are presented in Fig A2 in S1 Appendix. The coefficient of interaction is positive and statistically significant, indicating that the difference in the likelihood of choice of group contract between A vs B and A vs. C is higher by 15 percentage points in the case of absence of organizer than presence. This implies, from the predicted margins reported in Fig 2, that the presence of an organizer has relatively low influence on the choice of group contract in A vs. B (–6.8 percentage points) vis-à-vis A vs. C (7.0 percentage points). It is surprising to find that the presence of a middleman spurs the choice of group contract in A vs. C. Although it looks counterintuitive, we think that this results hold for contract C only where organizer role disappears that make the producer to think about the importance of organizers in the seed production contract. The price information treatment suggests that the choice of group contract is significantly higher (9 percentage points) in the concealed price information treatment than revealed case. Asymmetric information gives more opportunity for the seed organizer who enjoys high bargaining power in the seed contract, to leverage from ex post discretionary latitutde [48] (Wu, 2014), and it is quite expected that increased bargaining power through group contracts enable smallholder producers to intensely bargain and minimize the middleman exploitation.

As the choice of contract decision was made over five rounds, we controlled for the decision made in the previous round to capture producers' dynamic preferences. The results suggest that the choice of group contract in the previous round (*r-1*) does not significantly accelerate producers' preference for a group contract in the present round (*r*). Nevertheless, if the bargaining in the group yields a price gain over the offered price and leads to higher earnings in the previous round, this increases the likelihood of choosing a group contract in the current round.

Among the producers' socioeconomic characteristics, years of education has a negative effect on the choice of a group contract. This suggests that more educated producers might anticipate problems of coordinated efforts and free-riding in group contracts that bottle up

**Table 4. Determinants of producers' choice of the group contract.**

| Variables | (1) | (2) | (3) |
|---|---|---|---|
| | Dep. variable: choice of group contract (B or C) | | |
| *Treatment variables* | | | |
| Trt1: Contracts offered | -0.35*** | -0.35*** | -0.24*** |
| (A vs. C = 1 and A vs. B = 0) | (0.04) | (0.04) | (0.05) |
| Trt2: Presence of organizer | -0.10** | -0.10** | -0.10*** |
| | (0.04) | (0.04) | (0.04) |
| Interaction: Trt1 × Trt2 | 0.13*** | 0.13*** | 0.15*** |
| | (0.05) | (0.05) | (0.04) |
| Trt3: Price information in *r-1* | 0.09*** | 0.09*** | 0.07*** |
| (concealed = 1, revealed = 0) | (0.03) | (0.03) | (0.02) |
| Hybrid production | | -0.02 | -0.01 |
| | | (0.02) | (0.02) |
| *Previous round variables* | | | |
| Group contract (B/C) | | 0.08 | 0.06 |
| | | (0.07) | (0.06) |
| Group contract (B/C)* earnings | | 0.02*** | 0.02*** |
| | | (0.00) | (0.00) |
| Group contract (B/C)* price gain | | 0.04*** | 0.04*** |
| | | (0.01) | (0.01) |
| *Socioeconomic characteristics* | | | |
| Age | | | 0.00 |
| | | | (0.00) |
| Education | | | -0.01** |
| | | | (0.00) |
| Land owned | | | -0.01* |
| | | | (0.00) |
| *Seed contract experience* | | | |
| Proportion of land under seed production | | | -0.07* |
| | | | (0.04) |
| Years of experience in seed production | | | 0.00 |
| | | | (0.00) |
| Variety grown in last rabi season | | | 0.02 |
| (hybrid = 1, OPV = 0) | | | (0.03) |
| *Seed company associated with* | | | |
| Cooperative and public organization | | | 0.07** |
| | | | (0.03) |
| Private multinational | | | 0.02 |
| | | | (0.04) |
| Private national | | | -0.00 |
| | | | (0.03) |
| Round dummies | YES | YES | YES |
| Regional dummies | NO | YES | YES |
| No. of observations | 2,384 | 2,305 | 2,233 |

(*Continued*)

**Table 4.** (Continued)

| Variables | (1) | (2) | (3) |
|---|---|---|---|
| | Dep. variable: choice of group contract (B or C) | | |
| No. of producers | 596 | 593 | 575 |

Clustered standard errors in parentheses

*** p <0.01

** p <0.05

* p <0.1.

(The estimated standard errors are clustered at the individual level).

their preference. Furthermore, their individual performance is expected to be higher than that of non-literate farmers. Area of land owned has a negative effect on choice of group contract as by nature large farmers have higher bargaining power and scale of operations than small farmers. Furthermore, when producers allocate a large proportion of their land for seed production, an individual contract is preferred to a group contract (7 percentage points). Producers who have had experience with a seed production contract with cooperatives and public seed organizations are more likely to prefer group contracts than producers who have had a contract with local companies. To summarize, irrespective of the presence of organizer or not, the likelihood of choice of group contract B is significantly higher than for C, that is, the structural difference in the contract affects producers' preference although both B and C offer the option to bargain.

## Bargaining and price effects

Fairness in contracts involving smallholders is a big concern due to large differences in bargaining power and asymmetric bargaining process [49–52]. The distribution of surplus

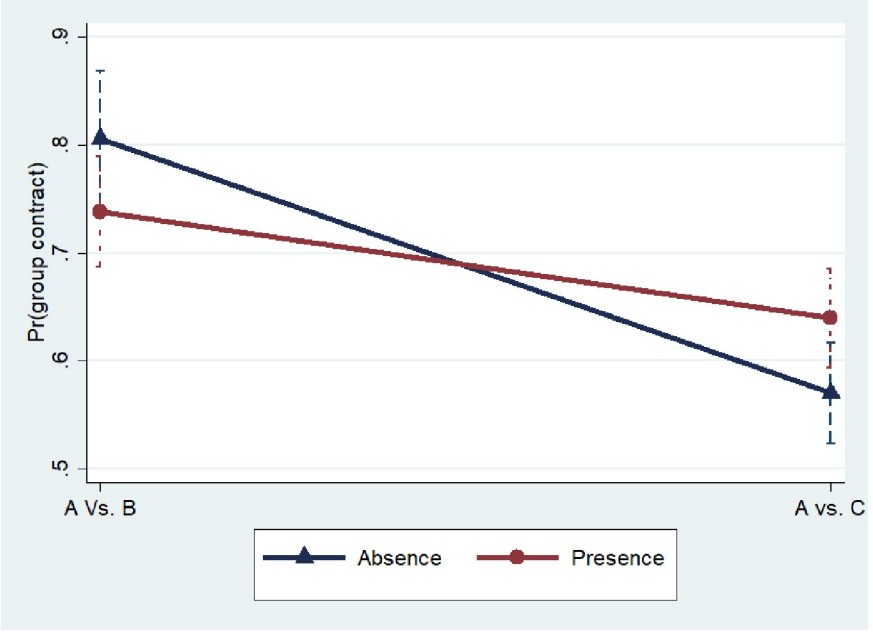

**Fig 2. Predicted probabilities of group choice.**

through allocation of bargaining power is examined by introducing bargaining between the SPG and organizer in contract B. In this section we report the results of bargaining on intended price gain (IPG) for producers as well as on intended price margin (IPM) for organizers (see Fig A1 in S1 Appendix). Intended price gain (IPG) is the maximum price gain that the producer group intends to obtain over and above the first price offered by the organizer, that is, IPG = $Price_{FB} - Price_{OB}$, where $Price_{FB}$ is the first price quoted by the SPG and $Price_{OB}$ is the first price offered by the organizer to the SPG in price negotiation in contract B. Actual price gain: APG = $-Price_B - Price_{OB}$, where $Price_B$ is the final price in contract B such that $Price_{FB} \geq Price_B$. Similarly, intended price margin (IPM) for the organizer is the maximum margin that an organizer intends to gain. (IPM = $Price_C - Price_{OB}$, where $Price_C$ is the company offered price). Actual price margin, APM = $Price_C - Price_B$. The average producers' gain (APG) is INR 1.6 per kg of seeds, which is reported in Fig 3. APG with reference to IPG in the concealed price treatment is 1 percentage point higher than in the revealed treatment, but the IPG of the concealed price treatment is higher (INR 0.86 per kg), that is, the overall price gain through bargaining is more effective in the concealed treatment (6.3 percentage points higher). Over the rounds, the price gain in the concealed treatment decreases from 41.7% in round 1 to 33.4% in round 5, with a dip in the middle round to 27.4%, whereas it increases in the revealed price treatment from 21.2% from round 1 to 34.5% in round 5, with a peak in round 3 at 40.4%.

The intended price margin for the organizer in both concealed and revealed price information is similar with IPM of INR 2.9 per kg (Fig 4). We observed a strategic behavior of the organizer by trying to increase the price margin by offering a lower price. Literature also reported the unequal bargaining power in agriculture contracts involving smallholders [49] and the use of discretionary adjustments in compensation and termination process, which secure the contractor high contract rent at the expense of smallholders [25,53,54]. The results indicated that although the ratio of APM to IPM is large in the initial rounds, over the rounds, due to producers intense bargaining the APM to IPM ratio reduced in both revealed and concealed treatments (37.93% and 46.42% of IPM in round 5 versus 40.74% and 51.72% in round 1,

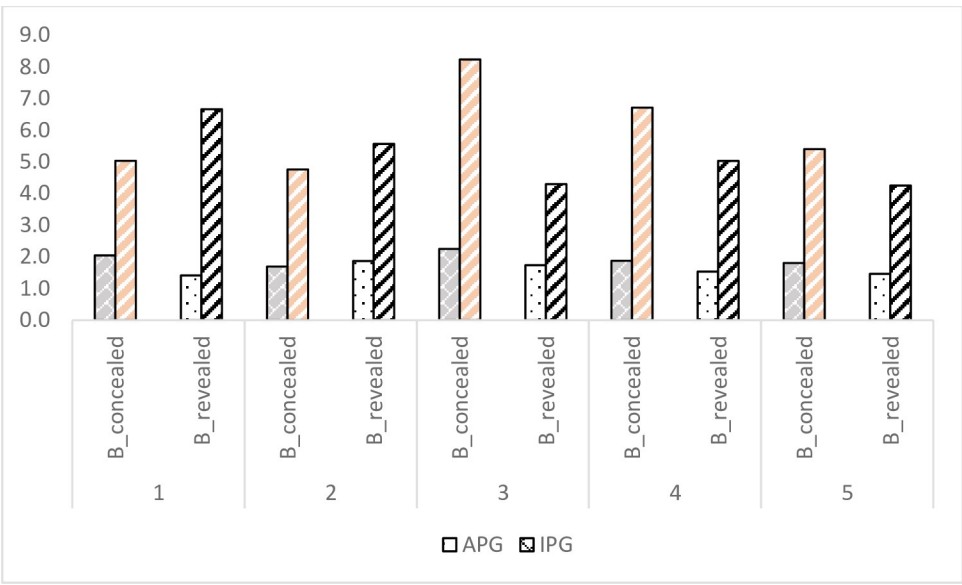

**Fig 3. Intended price gain (IPG) and average price gain (APG) in contract B.**

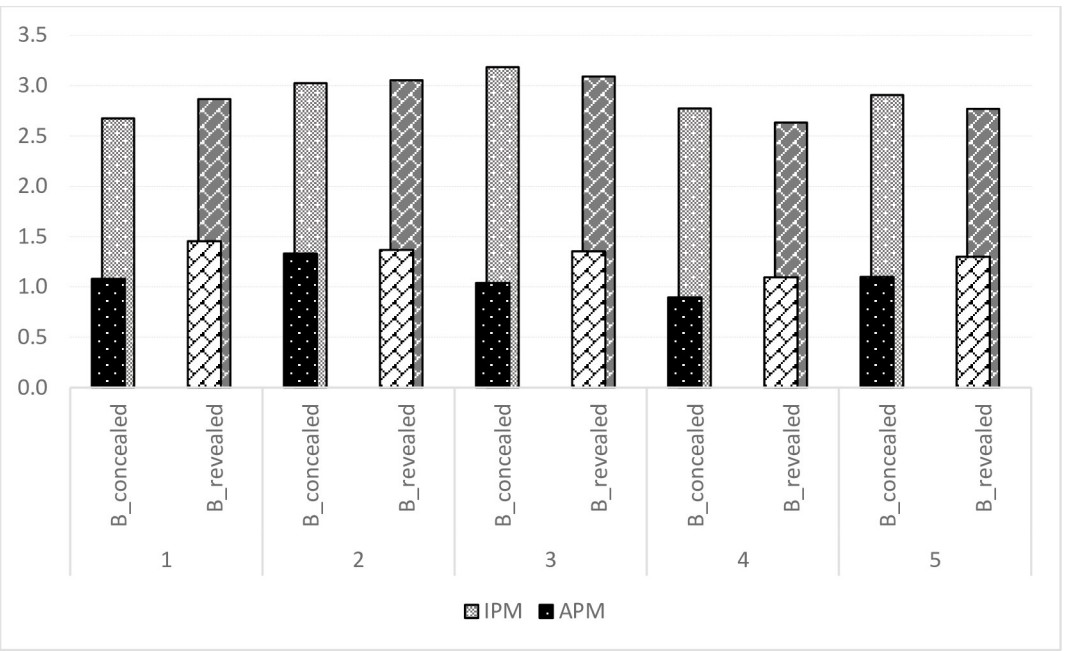

**Fig 4. Intended price margin (IPM) and average price margin (APM) in contract B.**

respectively, in concealed and revealed price treatments). The overall actual price margin is lower in the concealed case (37.93% of IPM) than in the revealed scenario (44.83% of IPM). In summary, bargaining is more pronounced in the concealed price treatment, resulting in better welfare distribution to farmers than in the revealed price treatment. The increased redistribution of contract rent, reaffirms the earlier findings that by improving the bargaining power of producers, the redistribution of surplus can be achieved [50,55,56].

## Dynamics of group formation

Compiling data at the village level, we derived three outcomes: group formation, Group break and group life. *Group formation* indicates whether the producers in the village coordinate among themselves to form an SPG in each round. *Group break* is the round at which the seed producers discontinue engaging in a group contract and *Group life* reports the number of rounds that a group sustains without any break. Of all the 60 villages we observed, seed producers formed an SPG in at least one of the rounds and 8 villages formed two groups in each village. Only 41% of the groups faced a break, of which 61% had regrouped in a later round. The groups which formed in later rounds continued to survive without any breaks. Also, the maximum breaks happened at round 3 (17.65%) and round 4 (13.24%). On average, SPGs survive nearly four rounds (average group life = 3.78) and 47% of the groups formed are sustained throughout five rounds without a break (see Table 5).

We estimated a conditional likelihood of group formation in the village (Eq 2) using a panel probit model and the marginal effects are presented in Table 6. The results indicate that group formation is higher in the villages where A vs. B contracts were offered than when A vs. C contracts were offered. The marginal effects suggest that group formation is 30 percentage points less likely when offered A vs. C vis-à-vis A vs B. The interaction effect suggests that group formation in the village does not significantly differ when offered different types of contracts with the presence and absence of a middleman, that is, the presence of an organizer

**Table 5. Group dynamics and sustainability.**

| Round | Group formation (%) (n = 68) | Group survival (%) (n = 68) | First break (%) (n = 28) | Reformation (%) (n = 17) | Second break (%) (n = 5) |
|---|---|---|---|---|---|
| 1 | 78 | 90 | - | - | - |
| 2 | 12 | 84 | 7 | - | - |
| 3 | 6 | 88 | 18 | 7 | - |
| 4 | 3 | 91 | 13 | 29 | 6 |
| 5 | 2 | 47 | 3 | 25 | 24 |
| Total no. of SPGs | 68 | 68 | 28 | 17 | 5 |

during the experiment does not affect group formation in the village. The dynamic variables such as having more producers and increased earnings from the group formed in the previous round increased the likelihood of the continuation of the group in the current round. The results indicate that membership strength and derived incentives from the group positively determine the group formed in the village. Among village characteristics, the villages with more multinational companies (MNCs) in operation and more hybrid producers have a higher likelihood of group formation. MNCs are mostly involved in hybrid seed production and demand higher production standards, that is, hybrid seed production requires stringent conditions, which necessitates high technical and managerial inputs, and the group is poised to address or facilitate attaining those high production standards and thereby reduce the probability of rejection. The villages having more national companies' operations are less likely to form a group. Those companies are mostly engaged in OPVs and are flexible in terms of seed production standards, with a fairly weak monitoring system. As some producers indicated, they do not find any difference in cultivation between normal grain production and seed production in such scenarios. Alternatively, producers can sell their produce at the grain market in the event of rejection or non-procurement with minimum cost implications. Further, the likelihood of group formation in villages decreases with an increase in the average landholding of producers. Often, land is considered as a proxy for wealth, and producers with more land implies more wealth; thus, producers may not desire cooperation with others and may express low willingness to form an SPG.

## Group survival

Using the group life outcome variable, we estimate the K-M survival function (Eq 3) to understand the survival rate of SPGs across different group contracts. Fig 5 presents the K-M survival curves for our sample. We find that the average survival rate of a group is 55%, implying that more than half of the SPGs formed are likely to survive throughout once they are formed. However, the groups that were formed in round one have a higher probability of surviving (67.9%), whereas survival is 12.5% for groups formed in round two. None of the groups survived until the last round if they were formed after round two. The survival rates of SPGs in contract B is significantly higher (68%) than for contract C (38%) ($\chi^2$ = 5.8; $p$ = 0.02). Further, SPGs formed in round one have a higher survival rate in the case of contract B (77%) than for contract C (53%). The survival rate of the group formed after round 2 is 17% in contract C and is zero for contract B. The survival rate goes to zero in both contracts if the groups are formed after the second round.

In order to understand the group survival rate, we further estimate the parametric Cox proportional hazard model and Table 7 presents the estimated parameters. The coefficients indicate that contract B has a significantly lower rate of hazard than contract C (hazard ratio [exp $(b_i)$] = 0.48). In other words, the group formed in contract B has two times more survival

**Table 6. Determinants of the formation of a seed producer group in the village.**

| Variables | (1) | (2) | (3) |
|---|---|---|---|
| | **Dep. variable group formed in village** | | |
| *Treatment variables* | | | |
| *Trt 1*: Contracts offered | -0.39*** | -0.30*** | -0.30*** |
| *(A vs. C = 1 and A vs. B = 0)* | (0.11) | (0.10) | (0.09) |
| *Trt 2*: Presence of middleman | -0.08 | -0.08 | 0.04 |
| | (0.12) | (0.11) | (0.13) |
| Interaction: *Trt 1 × Trt 2* | 0.12 | 0.12 | -0.03 |
| | (0.13) | (0.12) | (0.14) |
| *Trt 3*: Price information in *r-1* | 0.07 | 0.06 | 0.04 |
| (concealed = 1, revealed = 0) | (0.05) | (0.04) | (0.06) |
| *Lagged variables* | | | |
| Group formed in *r-1* | | -0.06 | -0.16 |
| | | (0.14) | (0.14) |
| Group formed * price gain *r-1* | | 0.04 | 0.03 |
| | | (0.03) | (0.03) |
| Group formed * earnings *r-1* | | 0.00 | 0.00* |
| | | (0.00) | (0.00) |
| No. of farmers in SPG *r-1* | | 0.03** | 0.04*** |
| | | (0.01) | (0.01) |
| SPG rejection *r-1* | | 0.03 | 0.11 |
| | | (0.10) | (0.10) |
| *Village characteristics* | | | |
| No. of MNCs in operation | | | 0.04** |
| | | | (0.02) |
| No. of national companies | | | -0.02* |
| | | | (0.01) |
| No. of cooperatives | | | -0.02 |
| | | | (0.03) |
| No. of local companies | | | 0.01 |
| | | | (0.01) |
| Seed production area | | | 0.00 |
| | | | (0.00) |
| No. of hybrid producers | | | 0.00** |
| | | | (0.00) |
| Proportion of SC caste | | | 0.00 |
| | | | (0.00) |
| *Average producers' characteristics in village* | | | |
| Age | | | 0.01 |
| | | | (0.01) |
| Education | | | -0.01 |
| | | | (0.02) |
| Land owned | | | -0.01*** |
| | | | (0.00) |
| Years of experience | | | 0.01 |
| | | | (0.01) |
| Round dummies | YES | YES | YES |
| Regional dummies | YES | YES | YES |

*(Continued)*

**Table 6.** (Continued)

| Variables | (1) | (2) | (3) |
|---|---|---|---|
| | Dep. variable group formed in village | | |
| No. of observations | 295 | 236 | 236 |
| No. of villages | 59 | 59 | 59 |

Standard errors in parentheses.

\*\*\* p <0.01

\*\* p <0.05

\* p<0.1.

probability than the group formed under contract C. The presence of an organizer significantly decreases (2.1 times) the survival rate of the group. When the price of seed is revealed to producers, producers do not find it attractive to continue to produce in the group and survival probability decreases by 2.2 times more than in the concealed price case. Further, the hazard rate decreases with an increase in group flexibility in terms of entry and exit of members. The survival rate of the group entry covariate is positive, but not statistically significant, whereas it is significant for group exit. The effects of size essentially imply that survival probability is 2.6 and 3.9 times higher, respectively, for groups that allowed new entry and exit for members than for groups that did not allow that. Entry into and exit from the group indicate the flexibility of the group to accommodate new members as well as respect individual preferences to exit the group without exerting group coercion. The survival rate of the price gain covariate is positive and statistically significant. It implies that a higher price gain through bargaining in the group would result in a higher survival rate (on average, 50% higher). In essence, the evidence suggests that the type of contract, price gain through bargaining, and group flexibility for entry

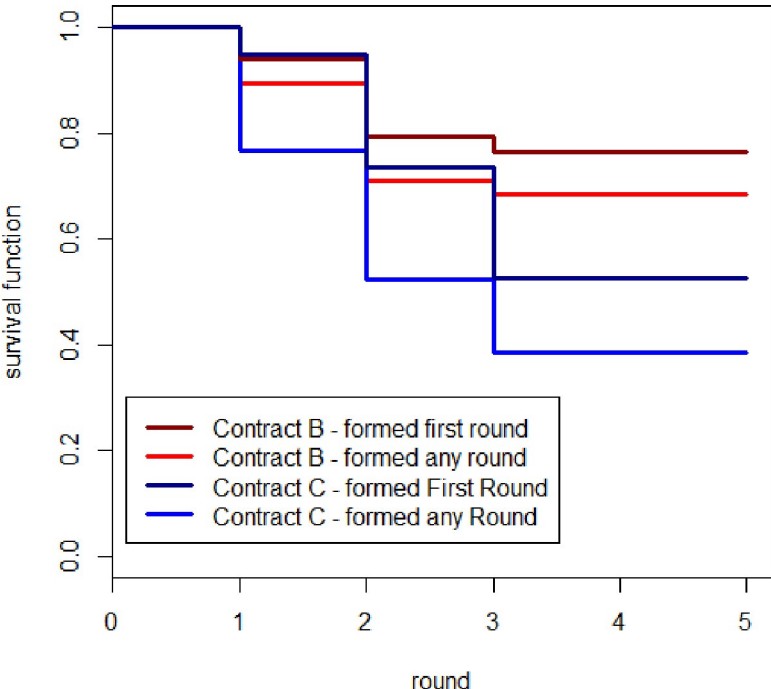

**Fig 5. K-M survival curve across contracts.**

**Table 7. Cox proportional hazard model for group survival (*n* = 68).**

| Variables | Coefficient (*b_i*) | Standard error | Confidence interval | Hazard ratio [exp (*b_i*)] |
|---|---|---|---|---|
| *Treatment variables* | | | | |
| Contract B | -0.728** | 0.379 | [0.230–1.016] | 0.483 |
| Presence of middleman | 0.763** | 0.398 | [0.983–4.682] | 2.145 |
| Price information (revealed = 1) | 0.799** | 0.423 | [0.971–5.090] | 2.223 |
| *Group dynamics* | | | | |
| Price gain | -0.413** | 0.211 | [0.437–1.001] | 0.661 |
| Group entry | -0.951 | 0.601 | [0.119–1.255] | 0.386 |
| Group exit | -1.356*** | 0.498 | [0.097–0.684] | 0.258 |

Concordance = 0.842 (se = 0.066), $R^2$ = 0.436 (max. possible = 0.969), Score (logrank) test = 38.17, $p < 0.01$

*** $p < 0.01$

** $p < 0.05$

* $p < 0.1$.

and exit improve the sustainability of the group, whereas the presence of a middleman and information asymmetry decrease the sustainability of the group.

## Conclusions

Contract farming in seed production has played an instrumental role in attracting private investment in seed research and production [57]. However, issues of the asymmetric bargaining and differential power relationship between contract parties across the value chain such as retail firms and producers and exploitation of producers by companies and middlemen have been identified [25,51,52]. In this regard, our study aims to analyze the changing power relationship between middlemen and producers through the introduction of a group contract, its impact on the welfare of actors and the group survivability. We conducted a lab-in-the-field experiment using rice seed producers and organizers. Two group contracts were proposed in the study, contract B (company → organizer → SPG) and contract C (company → SPG), with some improvisations to the current contract structure where producers were allowed to collectively bargain for a higher price. Producers were presented with the option to choose between the existing contract and one of the group contracts (either B or C).

The results showed that the producers prefer for group contract B involving organizer to the group contract C. As the contract B involve organizer in the contract that acts as a trust and familiarity factor and helps these smallholder producers to share the risks with organizer. Further, the coordination and advisory role of the organizer is well represented by this preference. In order to understand the extent of influence of the seed organizer on the choice of group contracts, we introduced a treatment in which a local organizer is present during the experiment while making the producer make a choice decision between contracts. In contrast to the expectations of middleman exploitation and rampant influence on collectivization of seed growers, we found that the presence of a local organizer does not affect the choice of a group contract. It is important to note that the existing contract farming arrangements have been well established in our study area for many years. Seed producers have more access to local organizers given their habitation in the village, and this proximity is a visible advantage for organizers. Thus, removing seed organizers from the system can be viewed as a revolutionary change. It is unlikely that farmers will adapt such a change given the risk and uncertainties involved in seed production, and the coordination and facilitation roles that organizers currently perform. A considerable amount of risk and coordination efforts is transferred to seed

organizers in the current system. Farmers view seed companies as an external entity with their own motives and thus they are reluctant to trust them without the presence of a local organizer. In our case, contract structure B, while allocating the benefits of improved bargaining power through group formation, does not alter the current system radically, but enjoys a redistribution of power and price margins from organizers to producers [50,56,58]. Although farmers' decision to form a group is not influenced by the presence of an organizer, group survival is negatively influenced by the presence of an organizer, that is, the middleman has an incentive not to have producer groups whereas producers acknowledge the role of a local organizer in the seed production contract. This contract is more acceptable to seed companies as well as they would continue to enjoy the current established contractual agreement with fewer organizers, which effectively reduces the transaction cost of individual contracts.

In contracts involving groups or cooperatives, one of the major concerns cited in the literature is group sustainability; more importantly when smallholder producers are tasked to coordinate efforts, that is, group dynamics and group sustenance are central when proposing a structural change in the seed contract. Groups formed survived with an average group life of 3.78 rounds. More than half of the groups (53%) that formed in the first round survived throughout the five rounds, indicating a very high sustainability. Our findings on group sustainability are a lower estimate given the random mix of producers in the experiment, which suggests that, in the real scenario, producers enjoy more flexibility to select among themselves those who share a higher degree of trust and interest to form the group, which further improves the sustainability of the group itself.

Our study suggests three insights for policymakers. First, revealing price information to producers helps to reduce price manipulation by organizers. Addressing this asymmetry of information is an option. Alternatively, by improving bargaining power of smallholders by allowing group contracts that spur the collective bargaining of producers, would achieve similar or better outcomes as that of revelation price information. Our study showed that the average price gain through bargaining in concealed price information (the status quo scenario) is higher than when the price is revealed. Second, removing the organizer from the seed value chain might increase the risks of producers and result in low-quality seeds, which ultimately reduces the preference for such contracts. The role of the seed organizer is viewed as a mix of traditional middleman and technical service provider or facilitator. Thus, the main objective of policy intervention is to distribute welfare to smallholder producers by addressing the concentration of bargaining power to the contractor, through group contracts and thereby minimizing middleman exploitation behaviour. This approach retains the role of organizer in coordination and technical service provision, and benefit from the prevailing trust of producers in organizers in the seed system. Third, contract B, which accommodates both organizers and producer groups, allows the production of good-quality seeds and attaining a higher price through collective bargaining. Allowing scope for collective bargaining through a producers' group in the contract can address both middleman exploitation and product output quality. Such an initiative improves group sustainability. Our study concludes that the type of contract, price gain through bargaining, and group flexibility for entry and exit improve the sustainability of the group, whereas information asymmetry decreases it.

We conclude the study highlighting few important limitations, which can be potential topics of further investigation. In this study, we explicitly did not account for free rider problem. The incentive for a member to free ride can have significance consequences on group performance and quality of the product which can cause net welfare loss for the member who practices free ride. Further, we assumed that the group introduces compliance mechanism when free rider problem exists beyond a threshold level. Second, there could be multiple ways that an organizer influences the group formation and performance. In the present study, the

organizers influence is captured only during the formation of group through their presence. But, though the transaction costs of organizer might reduce if engage in group, but the increased bargaining power of the group will eventually lead to better welfare distribution, reducing the organizer's profit. The influence of organizer on group performance and group break is a potential topic for future studies.

## Supporting information

**S1 Appendix.**
(DOCX)

## Author Contributions

**Conceptualization:** Prakashan Chellattan Veettil.

**Data curation:** Yashodha, Judit Johny.

**Formal analysis:** Prakashan Chellattan Veettil, Yashodha, Judit Johny.

**Funding acquisition:** Prakashan Chellattan Veettil.

**Investigation:** Prakashan Chellattan Veettil, Judit Johny.

**Methodology:** Prakashan Chellattan Veettil, Yashodha.

**Project administration:** Prakashan Chellattan Veettil, Judit Johny.

**Resources:** Prakashan Chellattan Veettil.

**Software:** Prakashan Chellattan Veettil.

**Supervision:** Prakashan Chellattan Veettil, Judit Johny.

**Validation:** Prakashan Chellattan Veettil, Yashodha.

**Visualization:** Prakashan Chellattan Veettil.

**Writing – original draft:** Prakashan Chellattan Veettil, Yashodha, Judit Johny.

**Writing – review & editing:** Prakashan Chellattan Veettil, Yashodha, Judit Johny.

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
