## [Decision Letter · Decision Letter 0]

7 Apr 2021

PONE-D-21-07085

Group contracts and sustainability: Experimental evidence from smallholder seed production

PLOS ONE

Dear Dr. Prakashan Chellattan Veettil,

Thank you for submitting your manuscript to PLOS ONE. After careful consideration, we feel that it has merit but does not fully meet PLOS ONE’s publication criteria as it currently stands. Therefore, we invite you to submit a revised version of the manuscript that addresses the points raised during the review process.

We look forward to receiving your revised manuscript.

Kind regards,

László VASA, PhD

Academic Editor

PLOS ONE

Journal Requirements:

Reviewers' comments:

Reviewer's Responses to Questions

**Comments to the Author**

1. Is the manuscript technically sound, and do the data support the conclusions?

Reviewer #1: Partly

Reviewer #2: Yes

2. Has the statistical analysis been performed appropriately and rigorously? 

Reviewer #1: Yes

Reviewer #2: Yes

3. Have the authors made all data underlying the findings in their manuscript fully available?

Reviewer #1: Yes

Reviewer #2: Yes

4. Is the manuscript presented in an intelligible fashion and written in standard English?

Reviewer #1: Yes

Reviewer #2: Yes

5. Review Comments to the Author

Reviewer #1: The paper's approach and overall quality is good. It is researching a remarkable piece of the field, a very actual problem in developing countries. I agree with te conclusions, accept the statistical toolset. I find it a bit long, but the journal's requirements do not limit it so I have to accept.

My advise to do changes as follows:

- a separate literature review should be created, extending the current resources;

- the limitations of the research should be indicated at the end of conclusions

Reviewer #2: This is a very well prepared paper on a topic of international relevance. The paper meets all the scientific requirements to be published in PLOSOne. The literature review, methods and results as well as discussion are very well elaborated.

6. PLOS authors have the option to publish the peer review history of their article (what does this mean?). If published, this will include your full peer review and any attached files.

Reviewer #1: No

Reviewer #2: No

---

## [Author Response · Author response to Decision Letter 0]

17 Jun 2021

We addressed the reviewer and editor comments and uploaded the details as a separate document, "Response to Reviewers"

---

## [Decision Letter · Decision Letter 1]

12 Jul 2021

Group contracts and sustainability: Experimental evidence from smallholder seed production

PONE-D-21-07085R1

Dear Dr. Prakashan Chellattan Veettil,

We’re pleased to inform you that your manuscript has been judged scientifically suitable for publication and will be formally accepted for publication once it meets all outstanding technical requirements.

Kind regards,

László Vasa, PhD

Academic Editor

PLOS ONE

Additional Editor Comments (optional):

Reviewers' comments:

Reviewer's Responses to Questions

**Comments to the Author**

1. If the authors have adequately addressed your comments raised in a previous round of review and you feel that this manuscript is now acceptable for publication, you may indicate that here to bypass the “Comments to the Author” section, enter your conflict of interest statement in the “Confidential to Editor” section, and submit your "Accept" recommendation.

Reviewer #1: All comments have been addressed

Reviewer #2: All comments have been addressed

2. Is the manuscript technically sound, and do the data support the conclusions?

Reviewer #1: Yes

Reviewer #2: Yes

3. Has the statistical analysis been performed appropriately and rigorously? 

Reviewer #1: Yes

Reviewer #2: Yes

4. Have the authors made all data underlying the findings in their manuscript fully available?

Reviewer #1: Yes

Reviewer #2: Yes

5. Is the manuscript presented in an intelligible fashion and written in standard English?

Reviewer #1: Yes

Reviewer #2: Yes

6. Review Comments to the Author

Reviewer #1: The authors made the recommended improvements so I can accept the paper for publication in its current form.

Reviewer #2: The authors have well revised their manuscript according to the reviewers' guidelines. All issues raised were handled.

7. PLOS authors have the option to publish the peer review history of their article (what does this mean?). If published, this will include your full peer review and any attached files.

Reviewer #1: No

Reviewer #2: No

---

## [Editor Report · Acceptance letter]

5 Aug 2021

PONE-D-21-07085R1 

*Group contracts and sustainability: Experimental evidence from smallholder seed production*

Dear Dr. Veettil:

I'm pleased to inform you that your manuscript has been deemed suitable for publication in PLOS ONE. Congratulations! Your manuscript is now with our production department. 

Kind regards, 

on behalf of

Prof. Dr. László Vasa 

Academic Editor

PLOS ONE